# Green Synthesis of Silver Nanoparticles from Anthocyanin Extracts of Peruvian Purple Potato INIA 328—*Kulli papa*

**DOI:** 10.3390/nano14131147

**Published:** 2024-07-04

**Authors:** Antony Alexander Neciosup-Puican, Luz Pérez-Tulich, Wiliam Trujillo, Carolina Parada-Quinayá

**Affiliations:** 1Bioengineering and Chemical Engineering Department, Universidad de Ingenieria y Tecnologia—UTEC, Lima 15063, Peru; neciosupantony@gmail.com (A.A.N.-P.); lperez@utec.edu.pe (L.P.-T.); 2Bioengineering Research Center—BIO, Universidad de Ingenieria y Tecnologia—UTEC, Lima 15063, Peru; 3Industrial Engineering Department, Universidad Tecnológica del Perú—UTP, Lima 15046, Peru; wtrujillo@warem.pe

**Keywords:** INIA 328—*Kulli papa*, nanoparticles, anthocyanin extract, green synthesis

## Abstract

In this work, AgNPs were synthesized using an anthocyanin extract from Peruvian purple potato INIA 328—*Kulli papa*. The anthocyanin extract was obtained through a conventional extraction with acidified ethanolic aqueous solvent. This extract acted as both a reducing and stabilizing agent for the reduction of silver ions. Optimization of synthesis parameters, including pH, reaction time, and silver nitrate (AgNO_3_) concentration, led to the optimal formation of AgNPs at pH 10, with a reaction time of 30 min and an AgNO_3_ concentration of 5 mM. Characterization techniques such as X-ray diffraction (XRD) and dynamic light scattering (DLS) revealed that the AgNPs had a crystallite size of 9.42 nm and a hydrodynamic diameter of 21.6 nm, with a zeta potential of −42.03 mV, indicating favorable colloidal stability. Fourier Transform Infrared (FTIR) analysis confirmed the presence of anthocyanin functional groups on the surface of the AgNPs, contributing to their stability. Furthermore, the bacterial activity of the AgNPs was evaluated by determining the minimum inhibitory concentration (MIC) and minimum bactericidal concentration (MBC). For *E. coli*, the MIC was 0.5 mM (0.05 mg/mL) and the MBC was 4.5 mM (0.49 mg/mL). Similarly, for *S. aureus*, the MIC was 0.5 mM (0.05 mg/mL) and the MBC was 4.0 mM (0.43 mg/mL). These results highlight the potential benefits of AgNPs synthesized from Peruvian purple potato anthocyanin extract, both in biomedical and environmental contexts.

## 1. Introduction

In recent years, silver nanoparticles (AgNPs) have become a focal point of extensive research, particularly within the biomedical sciences. Their distinct properties and versatile applications have ignited widespread interest and exploration in various disciplines such as medicine, biotechnology, and materials science [1,2,3]. However, modern nanoparticle synthesis methods often utilize chemicals with recognized environmental risk. Despite their efficacy in nanoparticle production, the ecological consequences of these chemicals cannot be overstated. Their discharge into ecosystems can result in persistent pollution, posing significant risks to aquatic and terrestrial life, while also potentially upsetting delicate ecological balances. It is imperative to prioritize the adoption of eco-friendly synthesis approaches to mitigate these adverse environmental impacts and ensure the preservation of our ecosystems [4,5,6]. Moreover, the need for costly purification steps may arise to ensure the thorough removal of residual chemicals. This necessity has spurred concerted efforts towards exploring alternative green routes for the synthesis of AgNPs. Among these alternatives, methods utilizing non-toxic chemicals, particularly those derived from plant extracts sourced from leaves, roots, fruits, or seeds, have emerged as particularly promising [7,8,9]. These approaches offer simplicity, safety, and cost-effectiveness, presenting viable solutions to mitigate environmental concerns associated with conventional synthesis methods.

Anthocyanins are natural pigments found in plant-based sources. Recently, their significance in nanoparticle synthesis, including silver, gold, and iron oxide nanoparticles, has attracted considerable attention due to their ability to act as reducing and stabilizing agents [10,11,12]. This reduction process involves the transfer of electrons from anthocyanins to metal ions, facilitating the formation of stable nanoparticles with controlled sizes and shapes [12]. Anthocyanins offer several advantages over conventional chemical agents, including biocompatibility, non-toxicity, and sustainability [13,14,15]. Furthermore, their abundance in plant sources makes them easily accessible and cost-effective for large-scale nanoparticle production.

The objective of this study is to extract anthocyanins from the Peruvian purple potato variety, INIA 328—*Kulli papa*, and subsequently identify the optimal conditions for synthesizing AgNPs. Additionally, the study aims to characterize the synthesized AgNPs and evaluate its antibacterial activity.

## 2. Materials and Methods

### 2.1. Materials

Peruvian Purple potato INIA 328—*Kulli papa* were purchased from a local farmers’ market in Jangas, Huaraz province, Ancash, Perú. Analytical grade silver nitrate (Merck, Darmstadt, DA, Germany, 99.9%), sodium hydroxide (Merck, ≥98%), ethanol (Baker Analyzed, Pennsylvania, PA, USA, 99.9%), citric acid (Merck, 99.5%), potassium chloride (Merck, 99.9%), and sodium acetate trihydrate (Merck, 99.9%) were purchased commercially.

### 2.2. Extraction and Quantification of Anthocyanins

The method of extraction of anthocyanins from Peruvian purple potatoes was carried out according to previous studies with some modifications [16,17]. A mass of 10 g of purple potatoes were grated and immersed in 50 mL of an ethanol–water mixture (4:6 volume ratio). To keep the extracted anthocyanins in a stable form, the pH of the medium was adjusted to 2.0 using 3 g of citric acid. The extraction was performed at 40 °C for 4 h in an aluminum-covered conical flask. Then, the solution was filtered and centrifuged at 4000 rpm for 10 min to remove suspended solid particles. The resulting extract was concentrated to 10° brix using a rotary evaporator at 50 °C and 80 mbar (BUCHI R-100, Flawil, Switzerland). Finally, the concentrated extract was stored at 4 °C in absence of light. The total anthocyanin content was determined using the differential pH method, which bases on the structural changes in chemical forms of anthocyanin and absorbance measurements at pH 1.0 and 4.5 [18,19]. For this analysis, 1 mL of extract was diluted separately with 9 mL of potassium chloride 0.025 mol/L (buffer pH 1) and 9 mL of sodium acetate 0.4 mol/L (buffer pH 4.5). The total anthocyanin content was quantified and expressed as cyanidin-3-glucoside equivalents as in the following equation [20]:(1)$$Total\;anthocyanins\;contents\left(\frac{mg}{L}\right)=\frac{A\times MW\times FD\times 1000}{\epsilon \times l}$$
where $A={\left(A_{vis-max}-A_{700}\right)}_{pH1}-{\left(A_{vis-max}-A_{700}\right)}_{pH4.5}$; MW is the molecular weight of cyanidin 3-O-glucoside (449.2 g/mol); FD is the extract dilution factor; ϵ is the molar extinction coefficient (26,900 L/mol∙cm); 1 is the path length in cm; and 1000 is the factor for conversion from gram to milligram.

### 2.3. Synthesis of Silver Nanoparticles

For the green synthesis of AgNPs, the anthocyanin extract was used as a reducing agent and silver nitrate (AgNO_3_) as a precursor [11,21,22,23]. For this propose, 50 mL of AgNO_3_ aqueous solutions were prepared and heated at 60 °C. Once the desired temperature was reached, 5 mL of anthocyanin extract (10 *v*/*v*%) was added. The formation of AgNPs was visually confirmed by a color change in the solution to yellow. A schematic representation of AgNPs synthesis is depicted in Figure 1.

To investigate the optimal conditions, different pH values (4, 7, and 10), silver ion concentrations (1 mM, 3 mM, 5 mM, and 7 mM) and reaction times (monitored at 5 min intervals) were tested. The effects of these parameters were analyzed using a UV-visible spectrophotometer.

#### 2.3.1. Effect of pH

To investigate the effect of pH on synthesis of AgNPs, the pH of the solution was adjusted to 4, 7, and 10 using 0.5 M sodium hydroxide. The pH of the reaction mixture was measured with a digital pH meter (OHAUS ST20, Pine Brook, NJ, USA). After that, the UV-visible spectrum was recorded in the range of 400–800 nm to monitor the formation of AgNPs and determine the optimal pH conditions for the synthesis. This optimal pH condition was then established for subsequent experiments to ensure consistent and efficient production of AgNPs.

#### 2.3.2. Effect of Reaction Time

The optimal reaction time for the complete reduction of silver ions was determined using a solution of 50 mL of AgNO_3_ (1 mM) and 5 mL of anthocyanin extract (10 *v*/*v*%), mixed at 60 °C. The pH of the reaction was adjusted to the optimized pH. The UV-visible spectrum was recorded every 5 min to monitor the reaction progress. According to the results, the best time for AgNPs synthesis was determined.

#### 2.3.3. Effect of Silver Nitrate Concentration

To consider the effect of the concentration of the metal agent, solutions of 1 mM, 3 mM, 5 mM, and 7 mM silver nitrate were prepared and heated at 60 °C. Subsequently, 5 mL of anthocyanin extract was added, and the pH was adjusted to 10. The reaction time was 30 min. After that, the UV-visible spectrum was taken, and the best concentration was determined.

### 2.4. Characterization of AgNPs

#### 2.4.1. UV-Visible Spectroscopic Profile of Synthesized AgNPs

UV-visible spectra were recorded in the range between 400 and 800 nm using a UV-visible spectrophotometer (SHIMADZU 3600, Tokyo, Japan). Double-distilled water (ddH2O) was used as a blank to adjust the baseline. All the samples were diluted to have an absorbance less than 1.4 AU.

#### 2.4.2. Dynamic Light Scattering (DLS) and Zeta Potential (ζ)

The average size and dispersion of AgNPs were evaluated through dynamic light scattering measured with a NICOMP Nano Z3000 System. Measurement was done using a He–Ne laser at a wavelength of 632.8 nm with 4.0 mW at 23 °C. Furthermore, in the same instrument, the zeta potential method was used to measure the electrostatic potential of the surrounded AgNPs. To prepare the samples, they were carefully diluted in double-distilled water to achieve the appropriate concentration.

#### 2.4.3. FTIR Spectroscopy

To identify the probable functional groups involved in the reduction and stabilization of AgNPs, Peruvian purple potato anthocyanin extract and synthetized AgNPs were analyzed using an FTIR spectrophotometer (SHIMADZU IRTracer-100, Tokyo, Japan) equipped with an ATR (PIKE Technologies, Madison, WI, USA). The FTIR spectra were recorded at a resolution of 2 cm^−1^ in the range of 400–4000 cm^−1^.

#### 2.4.4. X-ray Diffraction

The structural properties of the AgNPs were analyzed by X-ray diffraction technique using a D8 Focus diffractometer (Bruker AXS, Karlsruhe, Germany), operated at 40 kV and 40 mA with Cu-Kα radiation (λ = 1.506 Å). The sample preparation followed the method reported by Selvakumar et al. [22]. First, the colloidal solution was centrifuged at 15,000 rpm for 30 min to obtain the wet powdered nanoparticles. These nanoparticles were thoroughly washed with double-distilled water, followed by a wash with 100% ethanol to remove any unwanted organic matter. The cleaned nanoparticles were then transferred to a Petri dish and dried in a hot air oven at 40 °C for 12 h.

#### 2.4.5. Antibacterial Activity of Silver Nanoparticle

Two bacterial strains, Gram-negative *Escherichia coli* BL21 (ATCC BAA-1025) and Gram-positive *Staphylococcus aureus* (ATCC 29213), were grown in Luria–Bertani (LB) broth at 37 °C in a EUROTECH orbital shaker model FS-50B, at 200 rpm/min, for 24 h. Serial dilutions of both strains were carried out in LB broth until obtaining a turbidity on the McFarland scale of 0.5 [24].

The minimum inhibitory concentration (MIC) is defined as the lowest concentration of AgNPs that does not show bacterial growth in microplates after 24 h of incubation at 37 °C and 200 rpm/min. Dilutions of the bacterial culture were made with different concentrations of AgNPs (4.5 mM, 4.0 mM, 3.5 mM, 3.0 mM, 2.5 mM, 2.0 mM, 1.5 mM, 1.0 mM, 0.5 mM), 1ug/mL of ciprofloxacin was used as a positive control for *E. coli* and 1.5 ug/mL vancomycin were used as positive control for *S. aureus*, and the bacterial culture was used without nanoparticles or antibiotics as a negative control. In turn, the bactericidal activity of the anthocyanin extract was evaluated. Contamination control of the AgNPs and the LB broth used in these experiments was also carried out. After 24 h, the optical density was read in ELISA microplate reader (iMark Bio-Rad, Hercules, CA, USA) at 460 nm.

The minimum bacterial concentration (MBC) is considered as the lowest concentration of AgNPs in which colonies do not form. For this, 50 μL of the microplates were taken and cultured on plates with Mueller–Hinton agar (MHA). They were left for 24 h at 37 °C in an incubator (INCUCELL V 55 ECO line, Brno, Czechia) and the number of colonies was counted using ImageJ software (v.1.54).

## 3. Results and Discussion

### 3.1. Anthocyanin Content

Figure 2 shows the spectra of Peruvian purple potatoes anthocyanin extract in buffers at pH 1.0 and 4.5. According to Figure 1, maximum absorption was reached at wavelength $\lambda_{vis-max}=527nm$. Using Equation (1), the total content of extracted anthocyanins was 75.73 mg/100 g of fresh weight (FW). These results agree with other studies. For instance, Šulc et al. [25] reported a similar content of 78 mg/100 g FW in color-fleshed pulp potatoes, while Chen et al. [26] obtained 93.64 mg/100 g FW in purple sweet potato.

### 3.2. Synthesis of Silver Nanoparticles

#### 3.2.1. Effect of pH on AgNPs Synthesis

AgNPs in a colloidal system exhibit a surface plasmon resonance (SPR) absorption peak located between 390 and 450 nm using UV-visible spectroscopy analysis. This characteristic is commonly used to confirm the formation of AgNPs [27,28]. Figure 3 shows the UV–visible spectrum of 1mM AgNO_3_ and AgNPs synthesized at different pHs.

The AgNO_3_ solutions did not show any absorbance peak in the range of 390–450 nm. On the other hand, the effect of varying pH in the synthesis of AgNPs using NaOH 0.5 M showed that no absorption peaks were observed at pH = 4 and pH = 7. However, when the pH increased from 7 to 10, an absorption band appeared at 402 nm, which indicates the formation of AgNPs. This behavior may be attributed to the ionization of the anthocyanin biomolecule in an alkaline environment, enhancing both its reducing and stabilizing capabilities and facilitating the growth of AgNPs [29,30]. Consequently, pH 10 was chosen as the optimum pH.

#### 3.2.2. Effect of Reaction Time AgNPs Synthesis

Figure 4a shows the progression of the kinetic reaction over time, where an observable increase in the intensity of the yellow coloration corresponds to the continuous formation of AgNPs. This kinetic response is further supported by the UV-visible spectra presented in Figure 4b. Initially, there is a notable increase in absorbance within the first 30 min, indicative of the rapid nucleation and growth of AgNPs, followed by a stabilization phase. Figure 4c indicates the position of the SPR peak, which goes from 396 nm at 5 min to 405 nm at 30 min. This shift of the SPR peak is associated with the agglomeration of the silver nuclei being stabilized at 30 min. As a result, 30 min was chosen as the optimal reaction time for the complete formation of AgNPs.

#### 3.2.3. Effect of Silver Nitrate Concentration

Figure 5a shows UV-visible spectra at different concentrations of AgNO_3_. Absorption maxima at 405 nm revealed a controlled shape and size of synthesized nanoparticles. When the concentration of silver nitrate increased, the position of the maximum absorption peak did not show a shift. However, the maximum absorbance of the spectrum increased, which is an indication of the formation of more AgNPs with similar size and shape. Based on these results, a 5 mM AgNO_3_ solution is found to be optimum to produce uniformly sized AgNPs, as shown in Figure 5b.

### 3.3. Characterization of AgNPs

#### 3.3.1. Dynamic Light Scattering (DLS) and Zeta Potential (ζ)

Typical DLS size distribution profiles is shown in Figure 6a. AgNPs produced with 5 mM silver nitrate and 10 *v*/*v*% anthocyanin extract at 60 °C for 30 min showed an average diameter of 21.6 nm. Additionally, the stability and surface charge of AgNPs were determined by ζ-zeta potential. Figure 6b shows that the synthesized AgNPs were negatively charged with a zeta potential of −42.03 mV.

This finding indicates that anthocyanin molecules imparted a negative charge on the surface of the AgNPs, utilizing their stabilizing capacity. The resulting negative surface charge contributes to good colloidal stability and effective dispersion in the medium. This is achieved through repulsive forces between the particles, which prevent agglomeration and ensure a stable suspension. Consequently, our results demonstrate the significant role of anthocyanin extract in stabilizing AgNPs, confirming its potential as an efficient and natural agent for nanoparticle synthesis.

#### 3.3.2. FTIR Spectroscopy

FTIR spectroscopy analysis was carried out to determine the functional groups of anthocyanin extracted involved in the green synthesis of AgNPs. Figure 7 presents the FTIR spectra of colloidal AgNPs with anthocyanin extract, alongside the spectra of the anthocyanin extract alone. The two spectra exhibit overall similarity; however, there are slight shifts in the positions of the absorption bands. These shifts indicate potential interactions between the functional groups of anthocyanin molecules and the AgNPs, suggesting that these groups play a role in stabilizing the AgNPs. According to Bharadwaj et al., these functional groups are located on the surface of AgNPs [31,32], contributing to the development of surface charge, which correlates with the zeta potential measurements.

The FTIR profile for the anthocyanin extract shows bands at 3338 cm^−1^, 2115 cm^−1^, 1642 cm^−1^, 1410 cm^−1^, and 590 cm^−1^, whereas the synthesized AgNPs with anthocyanin extract shows bands at 3295 cm^−1^, 1640 cm^−1^, and 430 cm^−1^. The absorbance bands located at 3338 cm^−1^ correspond to the stretching vibrations of hydroxyl functional groups (-OH) from the anthocyanin extract, which is a polyphenolic compound [33,34]. Hydroxyl groups are known for their ability to donate electrons and adhere to a surface acting as a reducing and stabilizing agent. In this study, the participation of hydroxyl groups in the formation of AgNPs is confirmed through the shift of the -OH vibration positioned at 3338 cm^−1^ to 3295 cm^−1^. Additionally, the bands at 1642 cm^−1^ and 1410 cm^−1^ correspond to the stretching vibration of the C=C aromatic ring [23,35]. The absorbance band at 1280 cm^−1^ is attributed to the stretching vibration of -CO groups [36,37]. The absorption at 590 cm^−1^ indicates the presence of aromatic rings [38].

#### 3.3.3. X-ray Diffraction

Figure 8 shows the X-ray diffraction pattern of the AgNPs synthesized using anthocyanins extracted from Peruvian purple potatoes. The diffraction patterns showed good agreement with JCPDS (no. 04-0783). The crystalline nature of the AgNPs was verified through the presence of peaks at 38.12°, 43.83°, 64.12°, and 77.87°, corresponding to the (111), (200), (220), and (311) planes, respectively. This suggests the absence of other phases, highlighting the stability of the nanoparticles under the preparation conditions.

These findings unequivocally confirm the face-centered cubic (FCC) crystalline structure of the AgNPs. Therefore, the XRD clearly shows the crystalline AgNPs formed by the complete reduction of Ag^+^ ions by the ethanolic aqueous solution of anthocyanins. The average crystalline size was 9.42 nm, which was calculated using the Scherer equation from the full width at the half maximum (FWHM) peak broadening of the high-intensity peak (111) of the XRD graph [39]:(2)$$D\left(nm\right)=\frac{k\lambda }{\beta cos\theta }$$
where D is the crystalline size (nm); k is the shape factor, which is equal to 0.94 for sphere particles; β is the full width of the diffraction line at half of the maximum intensity measured in radians; λ is the X-ray wavelength of Cu-Kα radiation = 0.1506 nm; and θ is the Bragg angle.

#### 3.3.4. Antibacterial Activity of Silver Nanoparticle

The minimum bactericidal concentration (MBC) of AgNPs against the *E. coli* was 4.5 mM (0.49 mg/mL) (Figure 9). In contrast, against the *S. aureus* strain, AgNPs showed an MBC of 4.0 mM (0.43 mg/mL), as depicted in Figure 10. Additionally, minimum inhibitory concentrations were determined, with both strains exhibiting concentrations of 0.5 mM (0.05 mg/mL), as detailed in Table 1.

The ANOVA test and subsequent post hoc Tukey test were conducted to assess significant differences among the studied groups. Remarkably significant variances were observed between the negative control and other treatments, while no significant distinctions were found between the positive control (*Ciprofloxacin* for *E. coli* and Vancomycin for *S. aureus*) and the treatments examined. This suggests that AgNPs and Peruvian purple potato anthocyanins (specifically from the INIA 328—*Kulli papa* variety) exhibit potent bactericidal activity within concentrations ranging from 1.0 to 4.5 mM, achieving an average colony forming unit count equivalent to that of ciprofloxacin for *E. coli* (zero CFU). Regarding *S. aureus*, our nanoparticles demonstrated a colony-forming unit count of two within concentrations ranging from 0.5 to 4.5 mM, in contrast to the zero colony-forming units obtained with vancomycin.

## 4. Conclusions

Using green synthesis, AgNPs were successfully synthesized from anthocyanins extracted from Peruvian purple potato INIA 328—*Kulli papa*. Optimal conditions of pH 10, reaction time 30 min, and 5 mM AgNO_3_ concentration were determined. Optical study revealed the presence of the SPR peak at 405 nm, confirming the formation of AgNPs. A particle size distribution centered at 21.6 nm was observed using the DLS technique. The XRD profile showed the cubic structure of the Ag crystal face-centered with an average crystallite size of 9.42 nm. Functional groups involved in reducing Ag^+^ ions to Ag^0^ were identified by FTIR spectrum. Additionally, the AgNPs exhibited bactericidal activity against *E. coli* and *S. aureus* strains, with a MIC of 0.5 mM (0.05 mg/mL) for both strains. This synthesis process underscores the potential of utilizing natural resources for eco-friendly nanoparticle production, offering insights into their structural, optical, and antibacterial properties.

## Figures and Tables

**Figure 1 nanomaterials-14-01147-f001:**
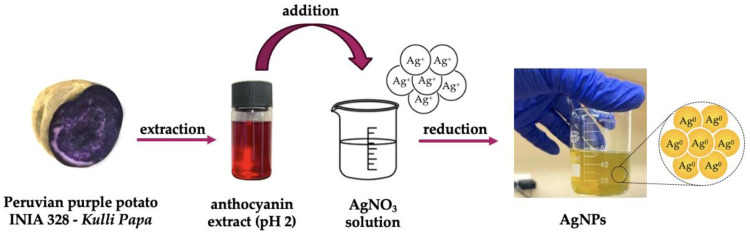
Schematic representation of the green synthesis of silver nanoparticles (AgNPs).

**Figure 2 nanomaterials-14-01147-f002:**
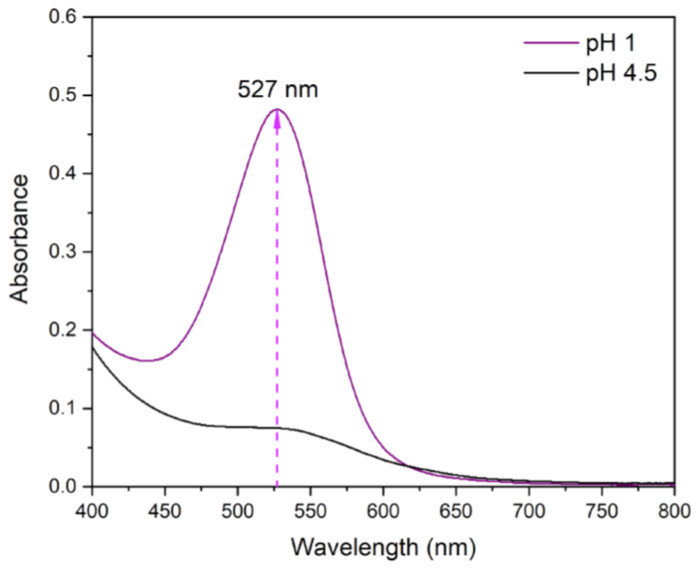
UV-visible spectrum of Peruvian purple potato anthocyanin extract at pH 1.0 and pH 4.5.

**Figure 3 nanomaterials-14-01147-f003:**
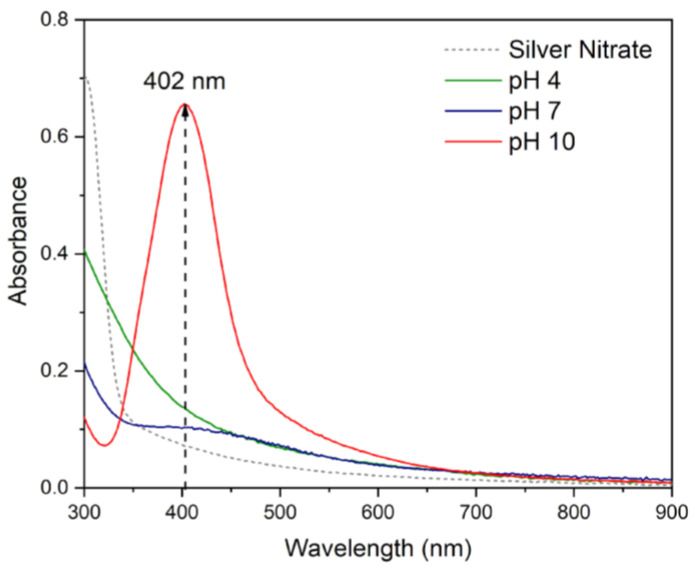
UV-visible absorption spectra of silver nitrate and AgNPs at different pHs.

**Figure 4 nanomaterials-14-01147-f004:**
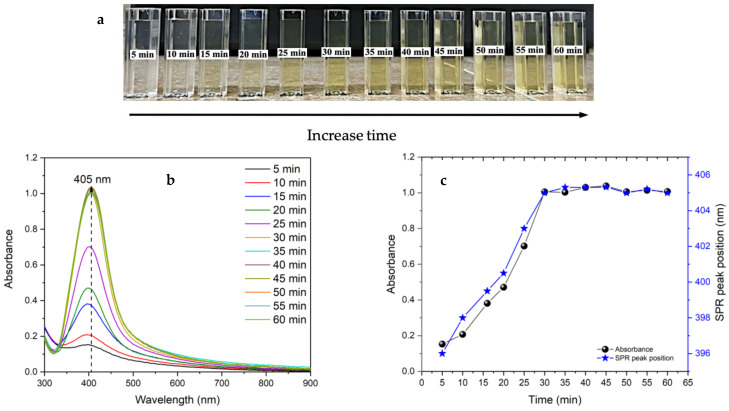
(**a**) Appearance of the AgNPs with increasing time, (**b**) UV-visible spectra of AgNPs at different time intervals, and (**c**) the evolution of absorbance and shift of the SPR peak.

**Figure 5 nanomaterials-14-01147-f005:**
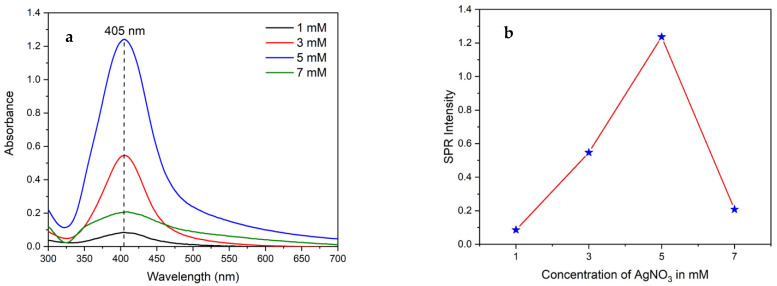
(**a**) UV-visible spectra of AgNPs synthetized at different AgNO_3_ concentrations and (**b**) the plot of maxima absorption against the concentration of AgNO_3_.

**Figure 6 nanomaterials-14-01147-f006:**
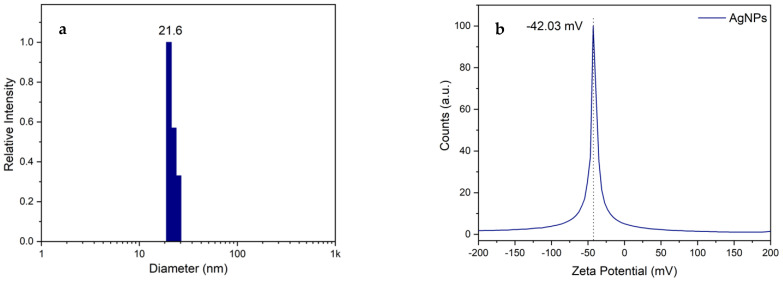
(**a**) Dynamic light scattering (DLS) and (**b**) zeta potential distribution of AgNPs.

**Figure 7 nanomaterials-14-01147-f007:**
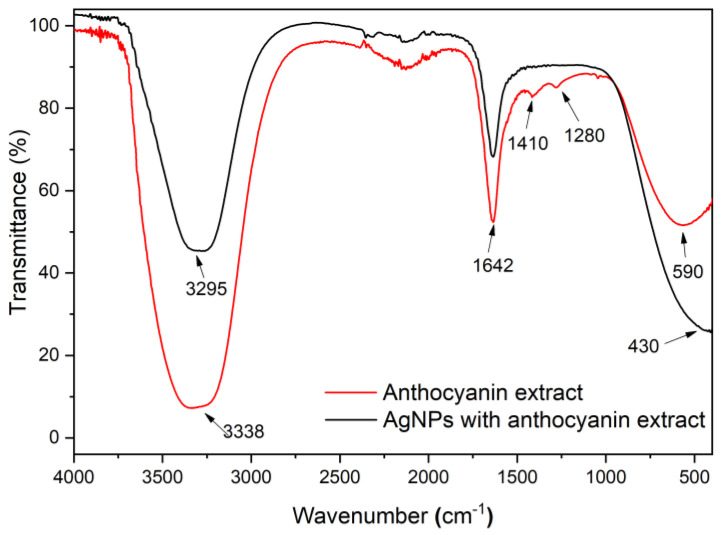
FTIR spectra of the anthocyanin extract and the AgNPs with anthocyanin extract.

**Figure 8 nanomaterials-14-01147-f008:**
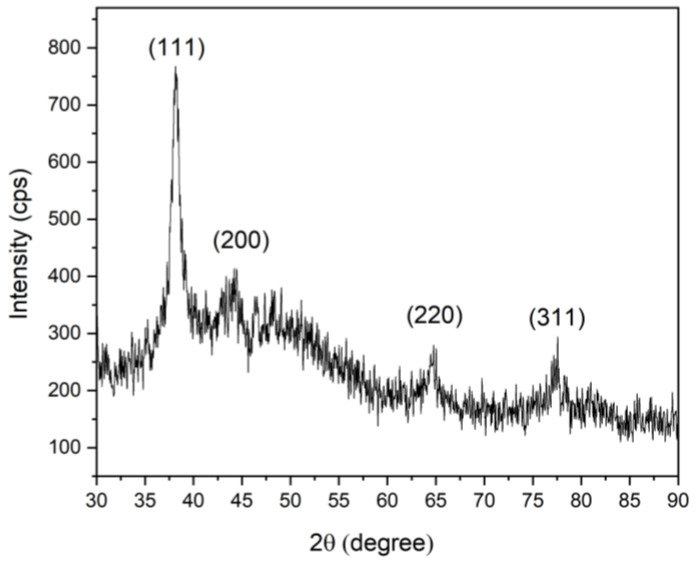
X-ray diffraction (XRD) pattern of synthesized AgNPs.

**Figure 9 nanomaterials-14-01147-f009:**
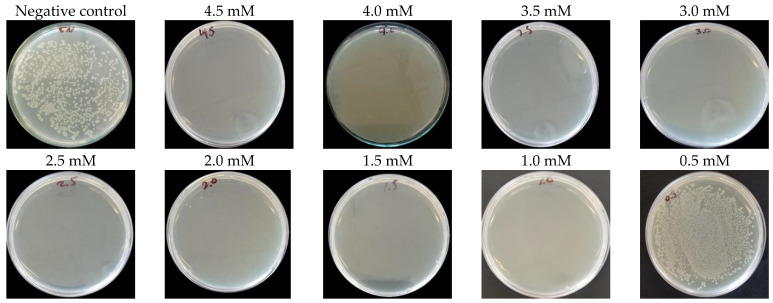
Minimum bactericidal concentration (MBC) of synthesized AgNPs against *E. coli* strain.

**Figure 10 nanomaterials-14-01147-f010:**
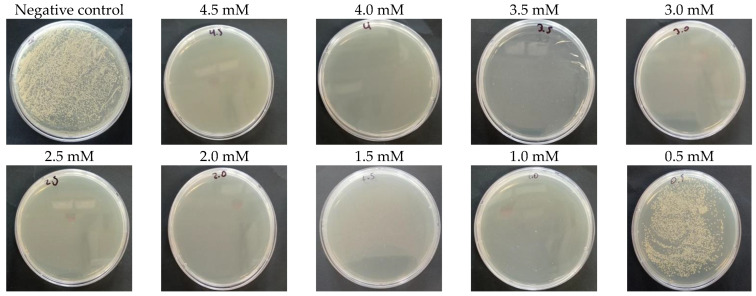
Minimum bactericidal concentration (MBC) of synthesized AgNPs against *S. aureus* strain.

**Table 1 nanomaterials-14-01147-t001:** MIC and MBC for *E. coli* and *S. aureus*.

Bacterial Strains	MIC (mM)	MBC (mM)
*Escherichia coli* BL21 ATCC BAA-1025	0.5	4.5
*Staphylococcus aureus* ATCC 29213	0.5	4.0

## Data Availability

Data are contained within the article.

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
