# Peer review of "Green Synthesis of Silver Nanoparticles from Anthocyanin Extracts of Peruvian Purple Potato INIA 328—Kulli papa"

_nanomaterials, 2024, doi:10.3390/nano14131147_

Round 1

Reviewer 1 Report

Comments and Suggestions for Authors

Line 297. The FTIR spectrum has the following label: with red: Anthocyanin extract

And in black: AgNPs,

This notation is not correct because the black line corresponds to the extract of the anthocyanins with the silver nanoparticles.

It is not indicated how the nanoparticles were prepared for analysis by XRD.

In the diffractogram on the Y axis only the u.a. legend is placed. Without any value, the correct thing is to graph the number of counts per second (value)

Although it cannot be reliably concluded that there are anthocyanin by FTIR and there is a great variety of these molecules, no analysis is carried out to determine the type of anthocyanin predominant in the potato used, based on this a possible mechanism and this should be consistent with the results obtained from z potential

Only the antibacterial potential of the nanoparticles obtained is realized and if one of the outstanding characteristics of anthicyanins is their antioxidant power, why were these types of tests not carried out to know this property characteristic?

The references used for this purpose are purple sweet potato, which is not the same as the potato used.

Only 5 references from 2022 are mentioned as the most recent, I consider that these should be updated and the number increased

Reviewer 2 Report

Comments and Suggestions for Authors

The article titled "Green synthesis of silver nanoparticles from anthocyanin extracts of Peruvian purple potato INIA 328 - Kulli Papa" authored by Alexander Neciosup-Puican et al. presents a novel approach to the biosynthesis of silver nanoparticles (AgNPs) using anthocyanin extracts. However, despite the innovative angle, several key aspects require substantial revisions and additional clarifications to enhance the manuscript's scientific rigor and contextual relevance.

Firstly, the introduction lacks a comprehensive review of the literature on the use of anthocyanins in the synthesis of AgNPs. Given the critical role of anthocyanins as potential reducing agents in green synthesis, the authors need to discuss previous studies in this area to position their work within the existing body of knowledge. This will not only validate the rationale behind using anthocyanins from purple potatoes but also highlight the novelty of their approach.

Concerning the methodology, the authors need to provide detailed information on the efficiency of the reduction process. This includes quantifying the conversion rate of Ag+ ions to AgNPs and discussing the kinetic aspects of the reaction. Additionally, the methods used to ensure the purity of the nanoparticles must be clarified. The control of unreacted silver ions within the synthesis environment is crucial for assessing the quality and applicability of the nanoparticles.

The possibility of chloride contamination in the extracts should also be addressed. Since chlorides can influence the morphology and stability of silver nanoparticles, the authors must discuss whether they have tested for and controlled the presence of chlorides in the anthocyanin extracts.

About dynamic light scattering analysis, the manuscript should specify the buffer or solution used during measurements. The choice of dispersant can impact the hydrodynamic size distribution and stability of nanoparticles; hence, justifying this choice will strengthen the validity of the DLS results. Moreover, a discussion on how this choice correlates with the intended applications of the AgNPs would provide deeper insights into the design of the experiment.

The authors are also encouraged to discuss the implications of their X-ray diffraction analysis more thoroughly, especially in terms of distinguishing between silver nanoparticles and potential silver chloride formations. This discussion should reference specific peaks identified in the XRD patterns and how these correlate with existing literature, such as the study by reference (10.1016/j.cis.2020.102246), which provides comparative data on similar nanoparticle systems.

Finally, the manuscript would benefit significantly from a detailed discussion on the proposed mechanism of silver ion reduction by anthocyanins. This should include the role of anthocyanins as both a reducing and stabilizing agent, detailing the chemical interactions at play. Theoretical discussions or previously published work supporting the proposed mechanisms should be cited to substantiate the claims.

In summary, while the study presents a promising approach to synthesizing silver nanoparticles using a natural reducer, the authors need to address several critical aspects to solidify the scientific foundations of their work. By expanding on the literature review, clarifying methodological details, and elaborating on analytical results and mechanisms, the paper can significantly contribute to green nanotechnology. I recommend the major revision. 

Reviewer 3 Report

Comments and Suggestions for Authors

Comment to the authors

I proceeded to analyze the manuscript entitled:

Green synthesis of silver nanoparticles from anthocyanin extracts of Peruvian purple potato INIA 328 - Kulli Papa

written by:

Antony Alexander Neciosup-Puican, Luz Pérez-Tulich, Wiliam Trujillo and Carolina Parada Quinayá

The manuscript describes a procedure for synthesizing AgNPs using anthocyanin extract from Peruvian purple potato, anthocyanin that as both a reducing and stabilizing agent for the reduction of silver ions. The formation of AgNPs was made evident using UV-Visible spectroscopy to highlight the plasmon resonance (SPR). The zeta potential (ζ) was mwasured, as well, together with the diameter using DLS obtained was of − 42.03 mV, indicating favorable colloidal stability. The antibacterial activity of the AgNPs was evaluated by determining the Minimum Inhibitory Concentration (MIC) and Minimum BactericidalConcentration (MBC) using a macro broth dilution assay.

The topic is, in my opinion, interesting and with practical application. The figures are suggestive and support the statements. References are in decent amount and they indicate that the authors are well aware of what has been published on the subject they are writing about. The article is carefully written, using good English, in my opinion, The content of the article sustains the Conclusion.

Moving to details, I found a few parts that, in my opinion, require improvement and additional clarification, as indicated on each item, and they are mentioned below.

-The abstract is more a brief Methods and Conclusion,rather than an Abstract. Avoid particular details in abstract and leave the for the appropriate section where you describe your work. Rewrite the abstract.

-lines 146:147: “Measurement was done using a solid-state He–Ne laser at a wavelength of 632.8 nm with a 4.0 mW at 23 °C.”

He-Ne laser is not a solid state laser. Verify and correct.

-equation (1): give citation or explain how you got the equation, for materials scientists whoa are not chemists.

-lines 293:296: “Figure 7 shows FTIR spectrum of colloidal AgNPs and anthocyanin extract which are similar with a slight shift in the band positions. This similarity suggest that certain functional groups present in the anthocyanin extract reside on the surface of the synthesized AgNPs [23-24].”

Explain why you stated: “functional groups present in the anthocyanin extract reside on the surface of the synthesized AgNPs” In Figured I believe that the 3295 and 3338 is not quite accurate; more likely the peaks are at the same position, which can be seen at enlarging the figure. Did you used a software to identify the peaks? Explain in manuscript and also explain why you stated thet they reside on the surface, in the absence of the shift. The hydrodynamic diameter is much bigger than the crystalite size. Can this be the prove, together with the smaller shift?

Round 2

Reviewer 1 Report

Comments and Suggestions for Authors

I see that most of the observations made were addressed, this provides greater clarity to the research developed and improves the quality of the article.

Regarding the correction in XRD, NOT only did the legend have to be changed from ua to cps, the NUMBER of cps must be placed on the “y” axis. This will allow us to observe that the diffractogram is reliable and will improve the quality, just as it is essential and mandatory to place the two theta angles on the "x" axis. This is how the number of counts per second should be placed on the y-axis.

Reviewer 2 Report

Comments and Suggestions for Authors

The revised manuscript is ready for publication. 
